mechanical engineering/engineering geology

cyclic load, high confining pressure,
digital speckle correlation method,
localized deformation bound, energy evolution

**Author for correspondence:**
Jianhua Hu
e-mail: hujh21@csu.edu.cn

# Analysis of fracture deformation field and energy evolution of granite after high confining pressure cyclic load pre-damage

Dongjie Yang, Jianhua Hu, Guanping Wen and Pingping Zeng

School of Resources and Safety Engineering, Central South University, Changsha, People's Republic of China

  DY, 0000-0002-0178-2775; JH, 0000-0003-1293-6829; PZ, 0000-0003-0163-3523

Considering the recent developments of deep mining, investigating the rock properties under high ground stress periodic load is highly demanded. Studies show that these characteristics are important factors affecting the long-term steadiness of rock. However, the mechanical properties of rock mass without macro failure after cyclic load should be studied. In the present study, granite in a mine is considered as the research object. A rock pre-damage experiment is conducted with the same cycles under different confining pressures and constant cycle upper and lower limit loads. The pre-damaged rock sample is subjected to a uniaxial compression test, and a high-speed charge couple device camera is used to record the speckle field image of the sample surface during the whole loading process. The digital speckle techniques are used to analyse the image of the pre-damaged sample, the deformation field of the specimen surface, the displacement dislocation value of the localized deformation area and the deformation energy value of the specimen surface. The results show that for the same cycle times, the confining pressure is less than 80 MPa, which has a weakening effect on the rock's axial strength. As the confining pressure approaches 120 MPa, the pre-damaged rock uniaxial peak strength increases. The characteristics of displacement dislocation energy evolution of the localized deformation bound are divided into three stages (pre-peak stage, peak point and post-peak stage). After pre-damage under the same cycle times and different confining pressure conditions, the deformation field evolution of rock is relatively consistent.

# 1. Introduction

The mining of mineral deposits and the excavation of large underground chambers and other construction projects have been continued to the deep development. The stress environment experienced by the deep rock mass is becoming increasingly complex. It should be indicated that the deformation and energy dissipation of rock during the fracture are fundamental problems of rock mechanics. Numerous scholars have investigated the mechanical response of rocks under cyclic loading. Ge *et al.* [1] carried out real-time meso-CT experiments on the rock fatigue failure using special triaxial loading experimental equipment incorporating a CT machine. They established a threshold for the mesoscale rock fatigue damage. Moreover, they studied the mechanism that the stress value of the cyclic loading affects fatigue failure. Then, they presented a preliminary law for describing the microscopic fatigue damage expansion of the rock. From the perspective of energy dissipation, the law of energy evolution of rocks under cyclic loading is studied [2–5]. Feng *et al.* [3] proposed a new nonlinear creep damage mode. The test shows that the creep properties of the rocks are affected by the initial damage. Meng *et al.* [4] studied the energy evolution characteristics of the rock failure for different cyclic load experiments. It showed the rule of the energy cumulative and dissipation before the peak. Through the design of rock experiments under cyclic loading, the mechanical properties and damage evolution mechanism of rocks are analysed [6–14]. Bagde & Petros [6] designed the uniaxial cyclic load experiment of sandstone by using MTS816 equipment. The experiment found that the fatigue strength of sandstone was a positive correlation to the frequency of the dynamic load, while it is negatively correlated to the amplitude. Martin [8] analysed the granite cracks process considering its stress–strain distribution and then carried out the cyclic loading and unloading test of damage. It was found that the damage stress gradually decreases and eventually approaches the initiation crack stress. Bagde & Petros [9] analysed the effect of uniaxial load amplitude and frequency about the rock strength degradation and its deformation behaviour. It was found that micro-cracking is the main reason of fatigue failure. Bastian *et al.* [10] designed uniaxial and triaxial cyclic compression experiments on sandstone. Moreover, they investigated the variations of mechanical characteristics under its experiment condition. Ghazvinian [11] investigated different properties and damage degree of granite under uniaxial cyclic load. And established a correlation between crack stress threshold and fatigue mode. Zhao *et al.* [12] studied that the water content had a significant impact on the peak strength and elastic modulus. Zhou *et al.* [13] studied the properties of frozen loess under multi-stage triaxial compression. Chen [14] studied the influence of confining pressure on rock damage degree by X-ray and triaxial experiment.

Scholars have carried out investigations on rock deformation field and energy evolution. Ray *et al.* [15] designed a cyclic load experiment of the sandstone. The results showed when the cycle numbers increased and the uniaxial compressive strength (UCS) decreased. Liang *et al.* [16] discussed and analysed the salt rock properties under cyclic load conditions. Chen *et al.* [17], Song *et al.* [18] and Li *et al.* [19] used the holographic interference, speckle interference and digital speckle correlation method (DSCM) for measuring the deformation field on the rock surface during the loading process. Moreover, they analysed the non-uniform deformation process of rock materials and described different parameters such as the width of the localized band. In general, non-uniform deformation evolutions and local deformations of the rock lead to the ultimate failure of the rock [20–22].

In the reviewed investigations, reasonable results and significant achievements have been obtained. However, there are still some unsolved challenges. For deep rocks under complex stress environment such as cyclic stress, it only causes damage and deformation but does not cause macro damage. The mechanical properties of the damaged state of the rock after cyclic stress are directly correlated to the safety and stability of deep rock engineering (for example pillar). In the present study, granites are used as the sample, and MTS815 is used to pre-damage the granite samples under different confining pressure cyclic loads. Then, the uniaxial compression test is performed on the damaged sample. DSCM is used to monitor the whole process of deformation and destruction. In the experimental study, a high-speed camera is used to record the speckle image, and DSCM is used to calculate the characteristics of the deformation field evolution, displacement dislocation and energy evolution of the localized deformation bound. Accordingly, it is very important to understand damage, fracture and instability of the rock through the displacement dislocation law and energy varying of rock deformation bound.

# 2. Experiments

## 2.1. High confining pressure cyclic load pre-damage test

The purpose of this experiment is to study 10 cycles of loading and unloading with different confining pressure conditions and analyse the degree of the rock pre-damage. Cylindrical samples with dimensions

**Table 1.** Physical parameters of granite.

| group | confining pressure (MPa) | l (mm) | d (mm) | dry mass (g) | natural mass (g) | saturated mass (g) | saturated mass after load (g) |
|---|---|---|---|---|---|---|---|
| A1-1 | 50 | 100.32 | 48.88 | 491.30 | 491.83 | 492.99 | 493.43 |
| A1-2 | | 100.33 | 49.02 | 495.16 | 495.72 | 496.55 | 497.13 |
| A1-3 | | 100.41 | 49.04 | 495.89 | 496.45 | 497.30 | 497.64 |
| A2-1 | 80 | 100.67 | 49.11 | 495.25 | 495.79 | 496.58 | 497.16 |
| A2-2 | | 101.01 | 49.12 | 492.53 | 493.02 | 493.91 | 494.66 |
| A2-3 | | 100.30 | 48.91 | 491.73 | 492.17 | 493.03 | 494.18 |
| A3-1 | 100 | 100.67 | 49.11 | 495.25 | 495.79 | 496.58 | 497.20 |
| A3-2 | | 101.01 | 49.12 | 492.53 | 493.02 | 493.91 | 499.00 |
| A3-3 | | 100.30 | 48.91 | 491.73 | 492.17 | 493.03 | 497.62 |
| A4-1 | 120 | 100.48 | 48.99 | 493.44 | 493.97 | 495.06 | 495.45 |
| A4-2 | | 100.27 | 49.01 | 491.61 | 492.09 | 492.93 | 493.96 |
| A4-3 | | 100.65 | 49.05 | 494.47 | 495.10 | 495.94 | 496.52 |
| B1 | 0 | 100.50 | 49.12 | 492.24 | 492.74 | 493.65 | — |
| B2 | | 100.77 | 49.04 | 495.15 | 495.73 | 496.61 | — |
| B3 | | 99.45 | 49.17 | 491.03 | 491.57 | 492.4 | — |

of $50 \times 100$ mm are created from the granite. In this experiment, granite is divided into five groups. Each group has three granites. The experiment steps are as follows:

(1) Table 1 presents the basic physical parameters of group A1 (other groups: confining pressure 80 MPa, A2; confining pressure 100 MPa, A3; confining pressure 120 MPa, A4; confining pressure 0 MPa, B). Five groups of granites are vacuumed (time 4 h) and saturated (time 2 h) to ensure that the rock samples are completely saturated.
(2) Using the MTS815 to apply the load of $2$ kN s$^{-1}$, the UCS of rock samples in group B is measured. Then, the corresponding UCS and stress–strain curves are obtained. It is observed that the UCS are 162.57, 164.2 and 167.87 MPa, respectively. The corresponding average value is 164.88 MPa.
(3) Figure 1 shows that the cyclic loading mode (loading in the form of sine wave, frequency: 0.00357 Hz) is conducted on the other four granite samples. By considering the UCS of rock as its axial compressive strength, the rocks are cyclically loaded and unloaded 10 times under the above-mentioned conditions. It should be indicated that the loading and unloading rates are set to $2$ kN s$^{-1}$.

## 2.2. Deformation and failure experiment of pre-damaged granite

After performing pre-damage based on different confining pressure cyclic loads, the sample is loaded in a uniaxial testing machine. A charge couple device (CCD) high-speed camera with a resolution of 5 million pixels is used to collect the surface speckle image of the deformation and destruction of the pre-damaged saturated water sample until the end of the experiment. Finally, DSCM is used to calculate the surface displacement field and deformation field of the pre-damaged sample during the loading. Figure 2 illustrates the main process and equipment used in this experiment.

# 3. Results

## 3.1. Analysis of the pre-damage degree of high confining pressure cyclic load

Figure 3 shows the stress–strain distribution of the water-saturated granite for 10 cycles, with axial stress and confining pressure set to 150 and 50 MPa, respectively. It is observed that cyclic load curve and

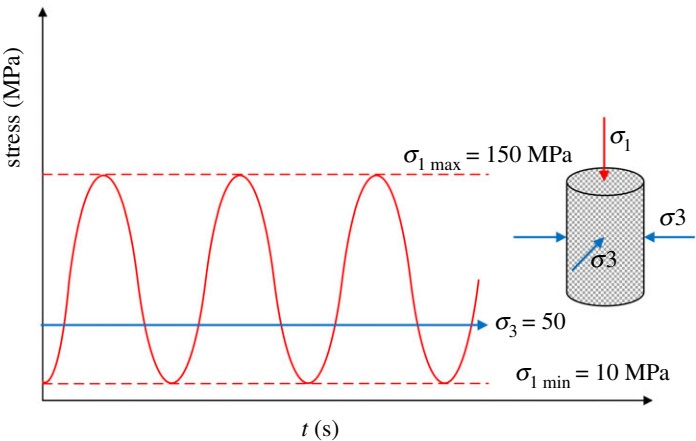

**Figure 1.** Cyclic loading scheme.

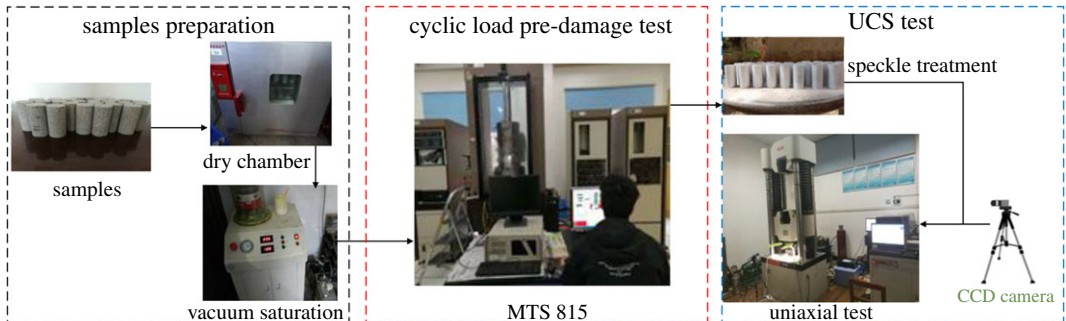

**Figure 2.** Experimental devices and process.

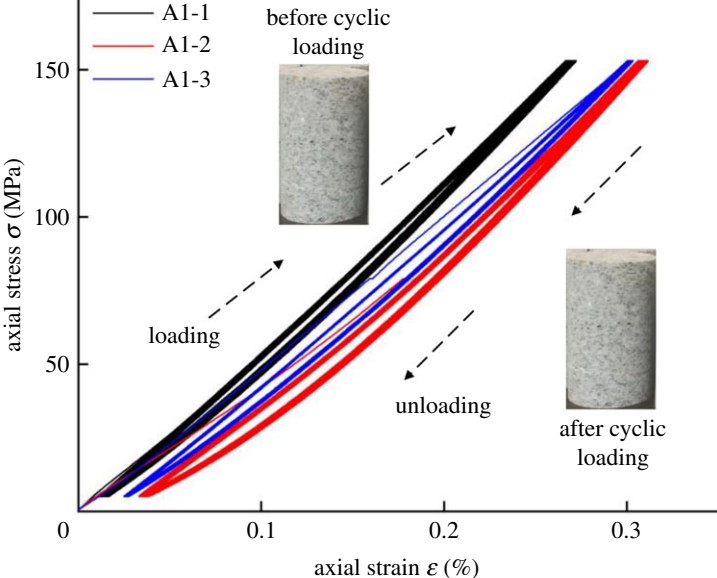

**Figure 3.** Stress–strain distributions for 10 cycles.

unload curve basically follow the stress–strain curve of the second. The first cycle has the largest hysteresis area. The area of hysteresis loop is basically unchanged from the second cycle to the tenth cycle. It can be concluded that the granites do not enter the plastic deformation stage after 10 cycles under confining pressure of 50 MPa. The results show that the minimum stress value of the unloading

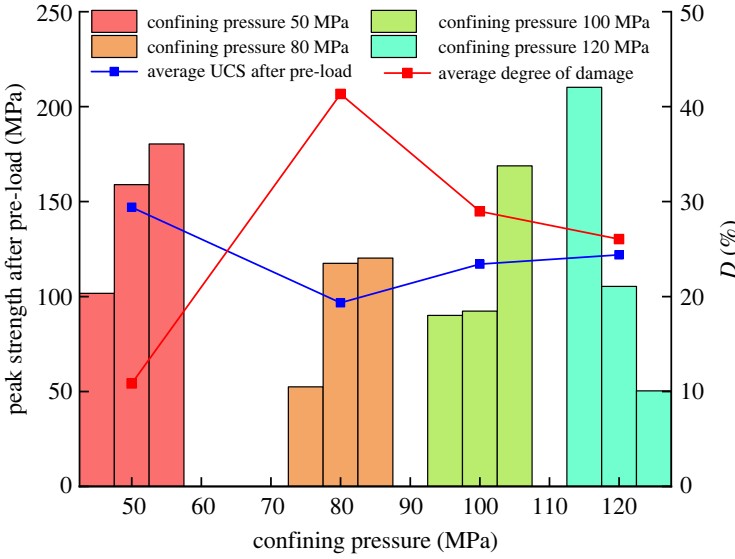

**Figure 4.** Peak strength and damage degree of the rock after pre-damage.

**Table 2.** Rock damage degree after 10 cycles of pre-damage.

| group | confining pressure (MPa) | peak strength after pre-load (MPa) | average peak strength after pre-load (MPa) | average degree of damage D (%) |
|---|---|---|---|---|
| A1-1 | 50 | 105.09 | 148.12 | 10.16 |
| A1-2 | | 158.95 | | |
| A1-3 | | 180.33 | | |
| A2-1 | 80 | 52.47 | 96.73 | 41.33 |
| A2-2 | | 117.49 | | |
| A2-3 | | 120.23 | | |
| A3-1 | 100 | 90.07 | 117.1 | 28.98 |
| A3-2 | | 92.37 | | |
| A3-3 | | 168.86 | | |
| A4-1 | 120 | 210.16 | 121.94 | 26.04 |
| A4-2 | | 105.32 | | |
| A4-3 | | 50.34 | | |

curve is lower than the elastic limit of this kind of granite. Moreover, the confining pressure enhances the UCS of granite.

Under this experimental condition with the same cycle number, when the confining pressures increase, the axial strain of the specimens do not increase or decrease regularly. It may indicate that high confining pressure leads to excessive increment of UCS. Therefore, the variation of the axial strain is not obvious under the condition of the same axial stress.

It is observed that the secondary peak strength of granite decreases after the pre-damage experiment with the increasing loading cycles. In other words, the rock damage degree continues to increase. The damage degree is defined as follows:

$$D = \frac{\sigma_c - \sigma_i}{\sigma_c}, \tag{3.1}$$

where $D$, $\sigma_c$ and $\sigma_i$ denote the degree of damage, the initial peak intensity and the secondary peak intensity, respectively.

Table 2 and figure 4 show the damage degree of the rock after 10 cycles of different confining pressure pre-damages. It is observed that the axial stress is 150 MPa. Moreover, it is found that for the same cycle

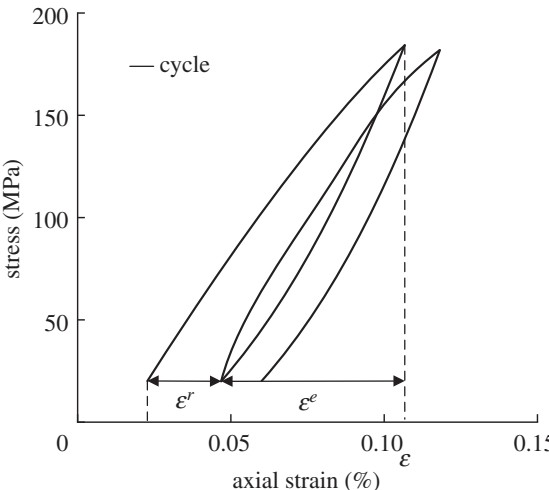

**Figure 5.** Cyclic loading of $\varepsilon^e$ and $\varepsilon^r$.

**Table 3.** Calculation results of elastic strain and residual strain (50 MPa confining pressure cycle for 10 times as an example).

| cycle | axial strain/$10^{-3}$ | | | circumferential strain/$10^{-3}$ | | |
|---|---|---|---|---|---|---|
| | $\varepsilon$ | $\varepsilon^r$ | $\varepsilon^e$ | $\varepsilon$ | $\varepsilon^r$ | $\varepsilon^e$ |
| 1 | 3.106 | 0.363 | 2.740 | 0.685 | 0.109 | 0.574 |
| 2 | 3.115 | 0.014 | 2.736 | 0.69 | 0.004 | 0.574 |
| 3 | 3.112 | −0.002 | 2.735 | 0.687 | −0.002 | 0.573 |
| 4 | 3.105 | −0.013 | 2.741 | 0.676 | 0.0007 | 0.562 |
| 5 | 3.093 | 0.0004 | 2.729 | 0.668 | −0.003 | 0.557 |
| 6 | 3.087 | −0.005 | 2.728 | 0.665 | −0.006 | 0.560 |
| 7 | 3.081 | −0.013 | 2.735 | 0.666 | −0.003 | 0.565 |
| 8 | 3.072 | −0.007 | 2.733 | 0.658 | −0.006 | 0.563 |
| 9 | 3.067 | −0.005 | 2.732 | 0.658 | −0.009 | 0.572 |
| 10 | 3.064 | 0.003 | 2.726 | 0.645 | −0.007 | 0.566 |

times, when the confining pressure is 80 MPa, the uniaxial peak strength of the pre-damaged granite decreases, while the corresponding degree of damage increases. The results indicate that the confining pressure is less than 80 MPa, which has a weakening effect on the rock's axial strength. As the confining pressure approaches 120 MPa, the pre-damaged rock uniaxial peak strength increases, while the degree of damage gradually decreases. It is observed that under this test condition, when the confining pressure is in the range of 80 to 120 MPa, the greater the confining pressure is, the smaller the damage to the rock is.

## 3.2. Strain evolution during cyclic loading

According to the stress–strain curve of cyclic loading, the $\varepsilon^e$ (represents elastic strain) and $\varepsilon^r$ (represents residual strain) during each loading were calculated. A cyclic loading process is shown in figure 5. The $\varepsilon^e$ and $\varepsilon^r$ are calculated and some data are shown in table 3. The $\varepsilon^e$ and $\varepsilon^r$ in the cyclic load process were normalized. Normalization was achieved by dividing the strain by the $\varepsilon^e$ and $\varepsilon^r$.

The relationship between the ratio of residual strain to strain and numbers of cycles is shown in figure 6. Under different confining pressures, $K_{ar}$ (the ratio between axial residual strain and axial strain) has an analogous trend to $K_{cr}$ (the ratio between circumferential residual strain and hoop strain). $K_{ar}$ and $K_{cr}$ first decrease with the cycles increased and then exhibit wave-like fluctuation. The phenomenon of a sudden drop in the first and second cycles indicates the natural pores inside the

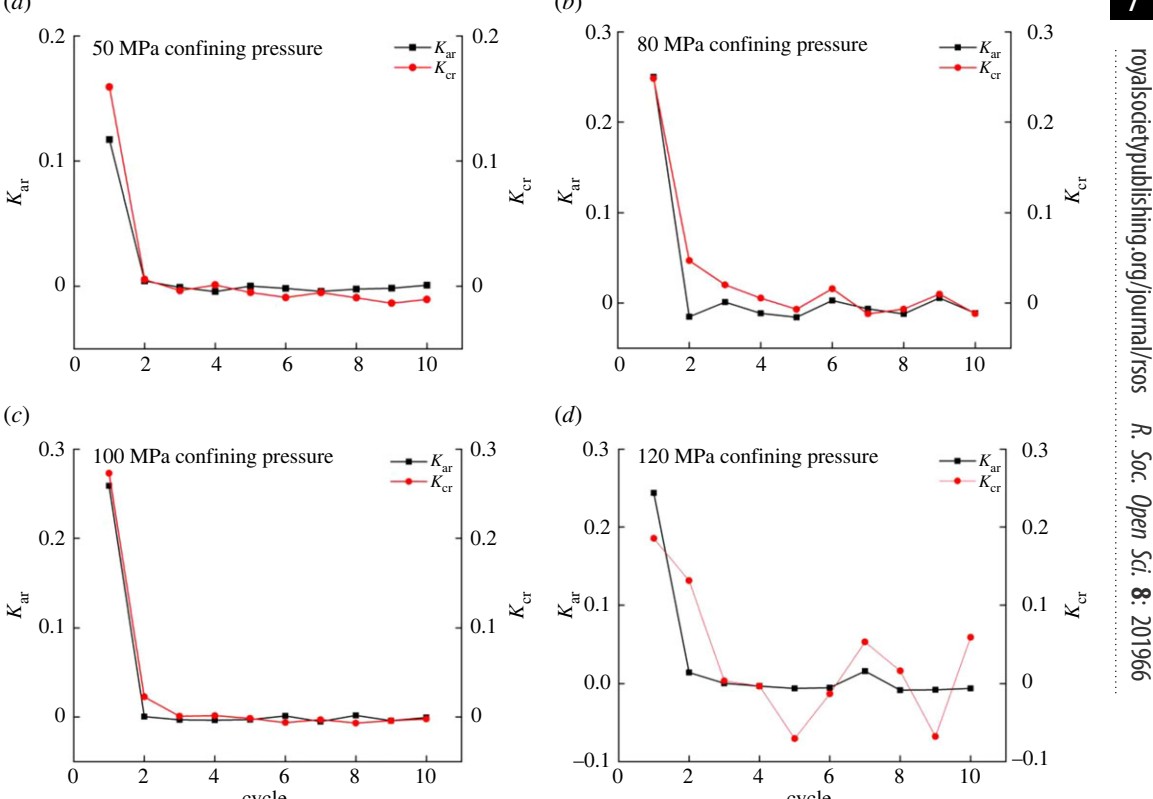

**Figure 6.** Variation trend of the ratio of residual strain to strain of rock.

rock are closed. The confining pressure increases to 120 MPa; $K_{cr}$ exhibits wave-like fluctuation. The magnitude of $K_{ar}$ relative to $K_{cr}$ gradually reduces, causing the rock to expand. The greater the pressure, the greater the effect on the hoop strain.

Figure 7 shows the change in the ratio between elastic strain and strain after normalization. When the confining pressure is 50 MPa, $E_a$ (the ratio between axial elastic strain and axial strain) has the same trend to $E_c$ (the ratio between circumferential elastic strain and hoop strain); that is, as the number of cycles increases, $E_a$ and $E_c$ gradually increase. But $E_a$ tends to be stable when the cycle number is about 8. When the confining pressure is 80, 100, 120 MPa, $E_a$ and $E_c$ have the same trend which increase in waves, and $E_a > E_c$. Under the experimental conditions, when the confining pressure is less than 80 MPa, the confining pressure weakens the strength of rock and increases the damage. When the confining pressure is greater than 80 MPa, the confining pressure strengthens the damage of rock and decreases the damage.

## 3.3. Evolution analysis of the deformation field of the pre-damaged specimen

Figure 8 shows the stress–strain evolution distribution of the pre-damaged specimen subjected to a cyclic load of 50 MPa for 10 times. In the presented distribution, the mark point 0 is used as a speckle reference image. Moreover, mark points 1–5 are used for deformed images. The evolution of the deformation field under different confining pressures with 10 cycles is calculated. The mark point 0–1 is located in the compression stage of the loading sample. The identification points 1–3 are located in the linear elastic stage of the specimen in the loading process. The evolution of the strain field shows that the sample is in the uniform deformation stage. Moreover, it is observed that the strain value at the moment corresponding to the mark point 3 is the highest. It may indicate that mark points 3–4, which are located in the plastic hardening stage of the sample loading, are also close to the peak point. Meanwhile, the strain field of the sample shows non-uniform characteristics, and the local deformation of the left and right sides of the sample is concentrated. Mark point 4 is located at the peak point. The localized deformation zone is formed. Mark points 4–5 are located in the plastic softening stage after the peak point of the sample loading. As the displacement of granite increases under the loading process, the stress decreases, and the change in the local deformation bound becomes more obvious. The localized deformation bound gradually elongates, expands and connects

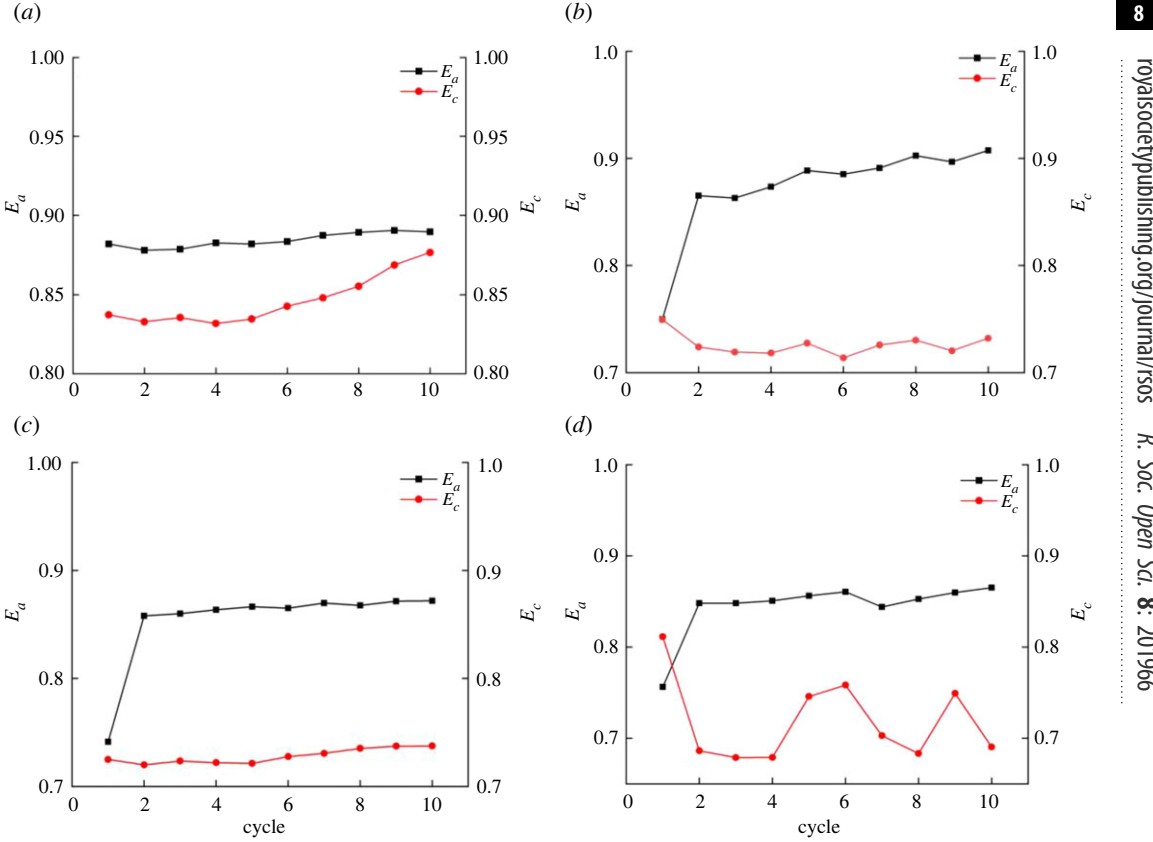

**Figure 7.** Trends of the ratio of elastic strain to strain under different confining pressures: (*a*) 50 MPa confining pressure, (*b*) 80 MPa confining pressure, (*c*) 100 MPa confining pressure and (*d*) 120 MPa confining pressure.

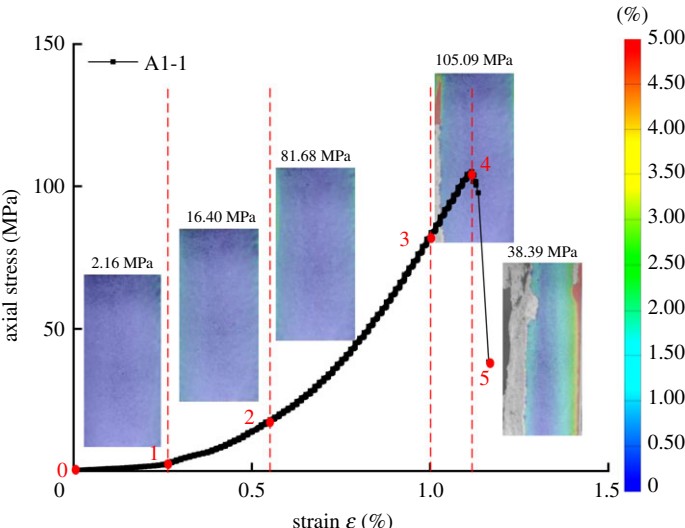

**Figure 8.** The strain field of rock at different loading stages.

until the specimen finally fails. Moreover, it is found that the deformation of the sample after the peak point is a dislocation slip deformation along the localized deformation bound.

The deformation field evolution process of the sample is obtained by processing the speckle image of the sample surface. Considering 0 in figure 8 as a reference point, the evolution law of the displacement and deformation field of the rock sample after pre-damage under high confining pressure cyclic load is observed at points 1, 2, 3 and 4. Table 4 shows the evolution process of the axial displacement (dy) and lateral displacement (dx) of rock after pre-damage under different confining pressure cyclic loads. This is consistent with the stage of natural granite during the loading process, while the failure process of the

**Table 4.** Evolution of the axial displacement and transverse displacement of the rock after pre-damage.

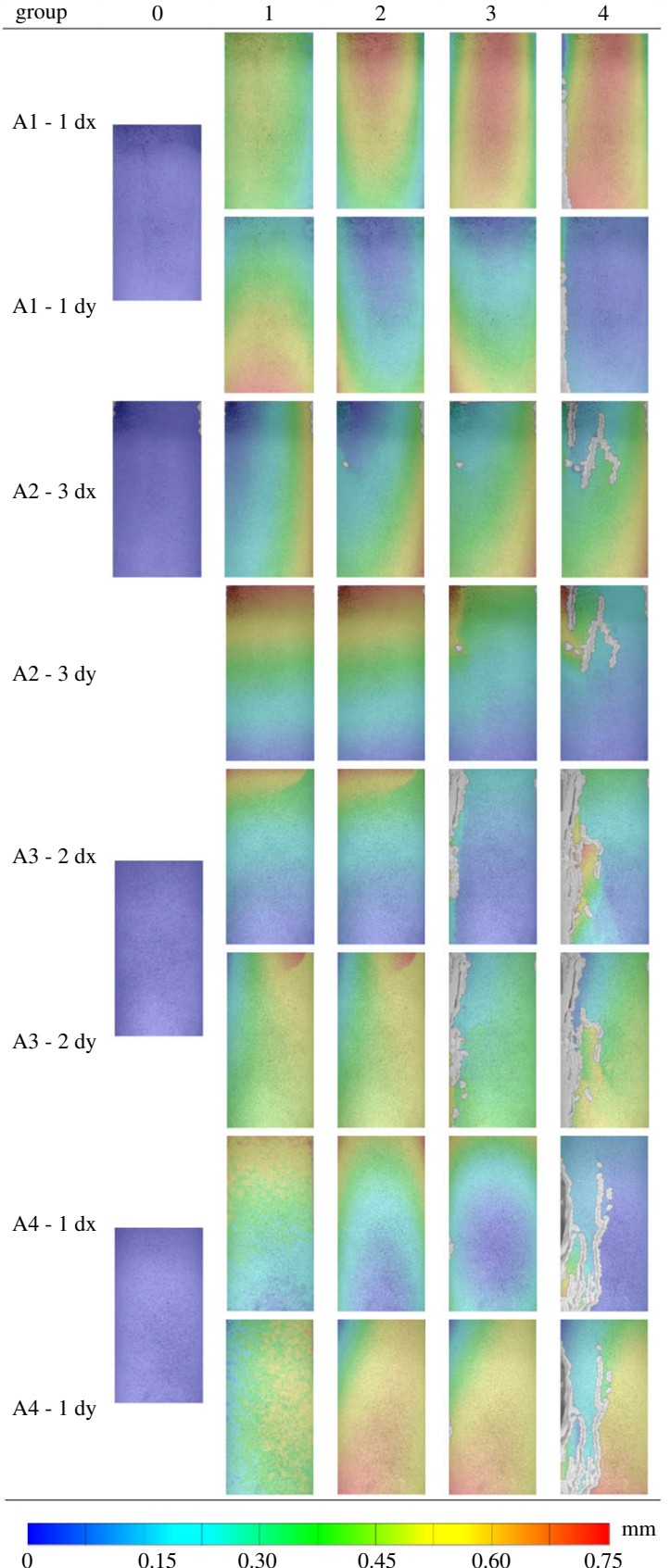

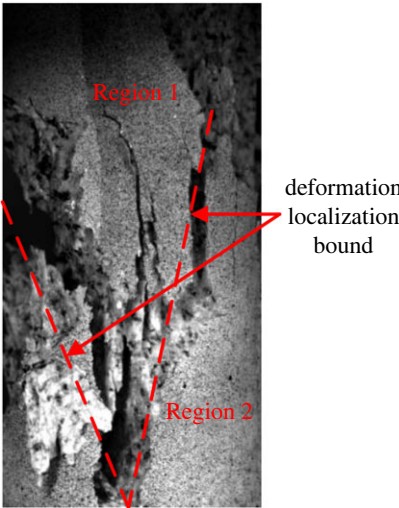

**Figure 9.** Schematic diagram of localized deformation bound identification.

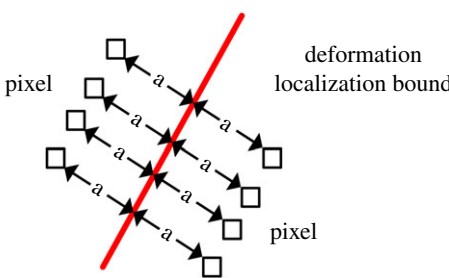

**Figure 10.** Diagram of displacement changes on both sides of the localized deformation bound.

pre-damaged rock samples is significantly different. The rule of the deformation shows as the confining pressure approaches 100 MPa, the local deformation stage appears earlier in the third position, and the degree of the localization increases, even macro cracks appear.

## 3.4. Dislocation of the pre-damaged specimen

In the present study, the division area of the localized deformation zone is divided according to the study of Shang *et al*. [23]. The localized area includes traces of fault, and the difference between the principal strains is more concentrated than the surrounding area. According to the final failure and the deformation field before the failure of the specimen, the localized deformation bound is identified in figure 9, where regions 1 and 2 are the regions outside the localized deformation bound. Figure 10 illustrates a schematic diagram of the displacement dislocation analysis on both sides of the localized deformation bound. The specific calculation method is as follows. First, the displacement field of each speckle image is calculated using the speckle processing software GOM. Then, four groups of pixels are selected symmetrically at $a = 2$ mm on both sides of the localized deformation bound. The difference in displacement is the displacement dislocation on both sides of the localized deformation bound. Finally, the method is used to calculate the displacement dislocation of the localized deformation bound during the whole loading process. Figure 8 illustrates the displacement dislocation, which is plotted as a curve corresponding to the stress and strain of the sample.

Figure 11 shows that under different confining pressure conditions, the displacement dislocation characteristics of the localized deformation bound are divided into three stages (pre-peak, peak point and post-peak). At the pre-peak stage, the displacement dislocation of the localized deformation bound maintains a uniform and linear change. It should be indicated that the dislocation occurs in the counter-clockwise direction. However, the degree of displacement dislocation is different for the samples with different pre-damaged confining pressures. At the peak point, the displacement dislocation evolution on both sides of the localized deformation bound moves in different directions, which in turn leads to the formation of macroscopic cracks and sample fractures. Based on the

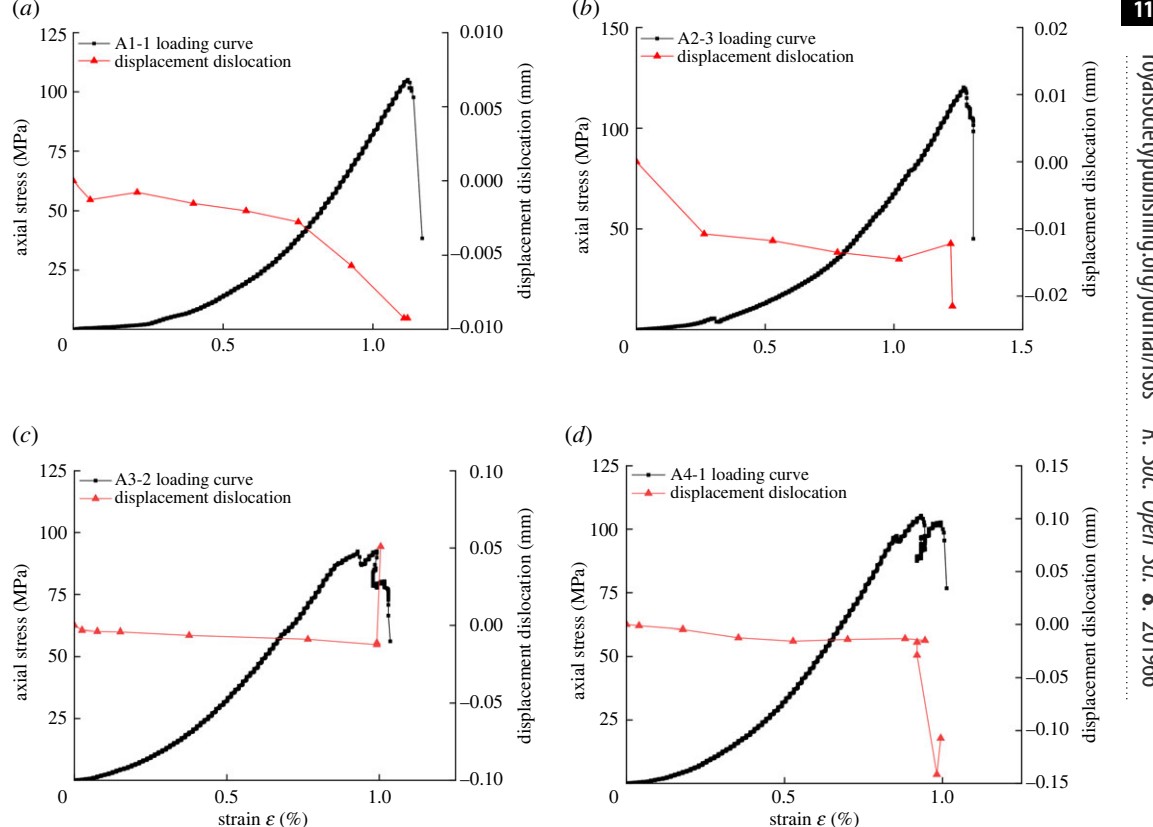

**Figure 11.** Evolution curves of displacement dislocation: (*a*) 50 MPa confining pressure, (*b*) 80 MPa confining pressure, (*c*) 100 MPa confining pressure and (*d*) 120 MPa confining pressure.

preliminary analysis, the displacement dislocation suddenly accelerates. This mainly originates from the formation, extend of micro-crack in the initial stage of the localized deformation zone. This may be the cause for the reduction of the overall capacity of granite. At the post-peak stage, the displacement of the localized bound changes. The first is that the dislocation of the sample end is large, while the change in the middle part of the sample is small, which is closely correlated to the degree of localization. The second is that the displacement of the sample undergoes a large change, which is consistent with the drop in stress after the sample peak.

## 3.5. Damage energy of the pre-damaged specimen

According to the evolution properties of the strain field before the rock failure, the localized deformation bound can basically divide the sample into two uniform deformation field regions $U_1$ and $U_2$. The average value of the strain at each point in the analysis area indicates the strain of the sample in the area. Moreover, according to the study of Zheng *et al.* [24], during the loading process of the sample, the area outside the localized deformation zone remains basically in the elastic state. Therefore, $U_1$ and $U_2$ can be calculated according to equation (3.2) and plotted together with the corresponding stress–strain distribution.

$$U = \frac{E}{2}(\varepsilon_1^2 + \varepsilon_2^2 - 2\mu\varepsilon_1\varepsilon_2)^2, \tag{3.2}$$

where $E$ and $\mu$ denote the elastic modulus and Poisson's ratio, $\varepsilon_1$, $\varepsilon_2$ are the principal strains on the surface of the rock specimen, and $U$ is the deformation energy density.

Figure 12 shows that the distribution of the deformation energy evolution curve can be divided into three stages (pre-peak, peak point and post-peak). Under the pre-peak period, the energy changes of uniform deformation fields $U_1$ and $U_2$ in the initial loading stage in figure 12*a*–*d* are mainly energy accumulation. Moreover, it is found that the characteristic curve of the energy evolution remains basically the same. In figure 12*a*, it is observed that the strain energy density curve begins to diverge when the strain is 0.578%, and the energy accumulation rate in the a-$U_2$ region begins to slow down.

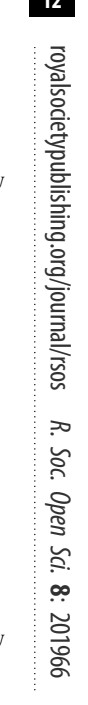

**Figure 12.** Evolution curves of deformation energy density: (*a*) 50 MPa confining pressure, (*b*) 80 MPa confining pressure, (*c*) 100 MPa confining pressure and (*d*) 120 MPa confining pressure.

In figure 12*b*, it is found that when the strain begins to change, the deformation energy density curve begins to diverge. However, the energy accumulation rate in the b-$U_1$ and b-$U_2$ regions remains basically unchanged until the rupture occurs. In figure 12*c*, the strain energy density curve begins to diverge when the strain is 0.376%, and the energy accumulation rate in the c-$U_2$ region exceeds the c-$U_1$ region. In figure 12*d*, the strain energy density curve begins to diverge when the strain is 0.670%, the energy accumulation in the d-$U_1$ region continues to accumulate, and the energy accumulation rate in the d-$U_2$ region begins to decrease. The main reason is the influence of the location and time of the micro-fracture of the localized deformation bound on energy accumulation.

At the peak point, a-$U_1$ and a-$U_2$ in figure 12*a* develop in opposite directions, and the energy change in the a-$U_1$ region remains basically the same as the pre-peak stage. Moreover, energy is continuously accumulated. In figure 12*b,c* and *d*, $U_1$ and $U_2$ release energy gradually at the peak point of loading.

In the post-peak period, the deformation energy density in figure 12*a*–*d* is similar to the evolution of the displacement dislocation of the localized deformation bound. The decrease of the axial stress of the specimen corresponds to the release of the deformation energy density in the region of $U_1$ and $U_2$. This indicates that the decrease of the axial stress of the rock is correlated to the failure of the localized deformation bound of the rock and causes the change of the energy outside the localized deformation bound. On the other hand, the change of the deformation energy in the post-peak period of the rock is regional, and the variation characteristics of different regions are different, which are mainly affected by the structure of the rock localized deformation bound.

# 4. Conclusion

In the present study, a pre-damage experiment is carried out on a rock sample during a high confining pressure cyclic loading. Then, a CCD camera is used for recording the uniaxial compression process of pre-damaged rock. DSCM is used to analyse the speckle image. Accordingly, the deformation field, displacement of the localized deformation zone and surface deformation energy value of rock failure are

obtained. The displacement dislocation evolution of the localized deformation zone under the load process is calculated. The obtained result shows that the linear evolution is basically maintained before the peak. Moreover, the peak point corresponds to the stage of the acceleration of the displacement dislocation of the localized deformation bound. The axial stress changes at the post-peak period of the rock are mainly affected by the displacement evolution of the localized deformation zone. According to the characteristics of the localized deformation zone, the outside of the localized deformation zone is divided into $U_1$ and $U_2$ uniform deformation field regions. Moreover, the evolution curve of the surface deformation energy density of the specimen in the compressive process is drawn. The analysis shows that energy release and energy accumulation of the rock during loading are correlated to the evolution of the localized zone and reflect the two forms of local energy release and overall energy release.

Ethics. This work having obtained permission from all the authors; we declare that the present experiments and manuscript were performed in accordance with the standard of academic conduct from Chinese academic society. All relevant ethical safeguards have been met in relation to patient or subject protection, or animal experimentation.
Data accessibility. All data, models and code generated or used during the study appear in the submitted article.
Authors' contributions. D.Y. was involved in methodology, formal analysis and writing the original draft. J.H. was involved in conceptualization, project administration and funding acquisition. G.W. was involved in the investigation. P.Z. was involved in the investigation.
Competing interests. We declare we have no competing interests.
Funding. Thanks for the National Key Research and Development Program of China (grant no. 2017YFC0602901) and the National Natural Science Foundation of China (grant no. 41672298).

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
