## [Peer Review File · Royal Society Open Science]

Review History

RSOS-201966.R0 (Original submission)

Review form: Reviewer 1

Is the manuscript scientifically sound in its present form?

Yes

Are the interpretations and conclusions justified by the results?

Yes

Is the language acceptable?

No

Do you have any ethical concerns with this paper?

No

Have you any concerns about statistical analyses in this paper?

No

Recommendation?

Accept with minor revision (please list in comments)

Comments to the Author(s)

This manuscript shows a comprehensive lab study to probe the energy characteristics of granite due to cyclic loading. The test is well designed and conducted and results of which can provide an in-depth understanding of the mechanical behaviour of rock, thereby providing guidance for underground rock engineering construction and applications. The manuscript is publishable prior to some improvements – please see the comments below.

- 1、 The third item "loading in the form of sine wave" in the test scheme, the specific parameters are unknown, which has a great impact on the experimental results, so the specific experimental parameters should be given.
- 2、 Fig 1 is not clear, please make it larger – It is a little bit hard for me to pick up key information.
- 3、 In the method section, it is unclear what is stress level applied on your sample in the cyclic load.
- 4、 Please modify the format of the formula in the text, such as in Formula 1 and 2 need to be italicized uniformly.

I think this paper is very interesting. If the above questions can be revised, I will recommend this paper to be published.

Review form: Reviewer 2**Is the manuscript scientifically sound in its present form?**

Yes

Are the interpretations and conclusions justified by the results?

Yes

Is the language acceptable?

Yes

Do you have any ethical concerns with this paper?

Yes

Have you any concerns about statistical analyses in this paper?

No

Recommendation?

Accept with minor revision (please list in comments)

Comments to the Author(s)

Overall, this is a good article. The pre-damaged rock sample is subjected to a uniaxial compression test, and a high-speed charge couple device camera is used to record the speckle field image of the sample surface during the whole loading process. The digital speckle techniques are utilized for analyzing the image of the pre-damaged sample, the deformation field of the specimen surface, the displacement dislocation value of the localized deformation area, and the deformation energy value of the specimen surface. In response to this manuscript, the following questions are raised, and the author is expected to reply.

1. In the section '2.1 High confining pressure cyclic load pre-damage test', the author is requested to complete the data for Groups A2, A3, A4 and B in the format of Group A1 in Table 1.
2. In the section '3.1 Analysis of the pre-damage degree of high confining pressure cyclic load', as can be seen in Table 2, the degree of damage of rock increases as the confining pressure increases from 50mpa to 80mpa; while the degree of damage of rock decreases as the confining pressure increases from 80mpa to 100mpa; and the degree of damage of rock remains essentially unchanged as the pressure increases from 100mpa to 120mpa, for which the authors are asked to briefly explain.
3. In the section '3.2 Strain evolution during cyclic loading', from the trend of the change of the ratio of elastic strain to strain under different confining pressure in Figure 7, it can be seen that when the confining pressure is 50Mpa, 80Mpa and 100Mpa separately, the ratio of circumferential elastic strain to circumferential strain has a more obvious linear relationship; however, when the confining pressure is increased to 120Mpa, the ratio of circumferential elastic strain to circumferential strain no longer has a linear relationship and fluctuates greatly, does this indicate that the rock has already undergone macroscopic breakage at this time, please briefly explain by the author.
4. In the section '3.4 Dislocation of the pre-damaged specimen', Figure 12 plots the deformation energy density evolution curve, which shows the process of energy release and energy accumulation of the specimen. When the energy accumulation reaches a maximum, a sudden release of energy occurs and the specimen is not undergoing energy accumulation, does it imply that the rock mass no longer has a bearing capacity at this point.
5. There are many verbose expressions in the paper, so the article content should be modified.

Decision letter (RSOS-201966.R0)

Dear Dr Yang

On behalf of the Editors, we are pleased to inform you that your Manuscript RSOS-201966 "Analysis of fracture deformation field and energy evolution of granite after high confining pressure cyclic load pre-damage" has been accepted for publication in Royal Society Open Science subject to minor revision in accordance with the referees' reports. Please find the referees' comments along with any feedback from the Editors below my signature.

Please submit your revised manuscript and required files (see below) no later than 7 days from today's (ie 28-Apr-2021) date. Note: the ScholarOne system will 'lock' if submission of the revision is attempted 7 or more days after the deadline. If you do not think you will be able to meet this deadline please contact the editorial office immediately.

Please note article processing charges apply to papers accepted for publication in Royal Society Open Science (<https://royalsocietypublishing.org/rsos/charges>). Charges will also apply to papers transferred to the journal from other Royal Society Publishing journals, as well as papers submitted as part of our collaboration with the Royal Society of Chemistry

(<https://royalsocietypublishing.org/rsos/chemistry>). Fee waivers are available but must be requested when you submit your revision (<https://royalsocietypublishing.org/rsos/waivers>).

on behalf of Prof R. Kerry Rowe (Subject Editor)
openscience@royalsociety.org

Reviewer comments to Author:

Reviewer: 1

Comments to the Author(s)

This manuscript shows a comprehensive lab study to probe the energy characteristics of granite due to cyclic loading. The test is well designed and conducted and results of which can provide an in-depth understanding of the mechanical behaviour of rock, thereby providing guidance for underground rock engineering construction and applications. The manuscript is publishable prior to some improvements – please see the comments below.

- 1、 The third item "loading in the form of sine wave" in the test scheme, the specific parameters are unknown, which has a great impact on the experimental results, so the specific experimental parameters should be given.
- 2、 Fig 1 is not clear, please make it larger – It is a little bit hard for me to pick up key information.
- 3、 In the method section, it is unclear what is stress level applied on your sample in the cyclic load.
- 4、 Please modify the format of the formula in the text, such as in Formula 1 and 2 need to be italicized uniformly.

I think this paper is very interesting. If the above questions can be revised, I will recommend this paper to be published.

Reviewer: 2

Comments to the Author(s)

Overall, this is a good article. The pre-damaged rock sample is subjected to a uniaxial compression test, and a high-speed charge couple device camera is used to record the speckle field image of the sample surface during the whole loading process. The digital speckle techniques are utilized for analyzing the image of the pre-damaged sample, the deformation field of the specimen surface, the displacement dislocation value of the localized deformation area, and the deformation energy value of the specimen surface. In response to this manuscript, the following questions are raised, and the author is expected to reply.

1. In the section '2.1 High confining pressure cyclic load pre-damage test', the author is requested to complete the data for Groups A2, A3, A4 and B in the format of Group A1 in Table 1.
2. In the section '3.1 Analysis of the pre-damage degree of high confining pressure cyclic load', as can be seen in Table 2, the degree of damage of rock increases as the confining pressure increases from 50mpa to 80mpa; while the degree of damage of rock decreases as the confining pressure increases from 80mpa to 100mpa; and the degree of damage of rock remains essentially

unchanged as the pressure increases from 100mpa to 120mpa, for which the authors are asked to briefly explain.

3. In the section '3.2 Strain evolution during cyclic loading', from the trend of the change of the ratio of elastic strain to strain under different confining pressure in Figure 7, it can be seen that when the confining pressure is 50Mpa, 80Mpa and 100Mpa separately, the ratio of circumferential elastic strain to circumferential strain has a more obvious linear relationship; however, when the confining pressure is increased to 120Mpa, the ratio of circumferential elastic strain to circumferential strain no longer has a linear relationship and fluctuates greatly, does this indicate that the rock has already undergone macroscopic breakage at this time, please briefly explain by the author.

4. In the section '3.4 Dislocation of the pre-damaged specimen', Figure 12 plots the deformation energy density evolution curve, which shows the process of energy release and energy accumulation of the specimen. When the energy accumulation reaches a maximum, a sudden release of energy occurs and the specimen is not undergoing energy accumulation, does it imply that the rock mass no longer has a bearing capacity at this point.

5. There are many verbose expressions in the paper, so the article content should be modified.

===PREPARING YOUR MANUSCRIPT===

===PREPARING YOUR REVISION IN SCHOLARONE===

Author's Response to Decision Letter for (RSOS-201966.R0)

See Appendix A.

Decision letter (RSOS-201966.R1)

Dear Dr Yang,

It is a pleasure to accept your manuscript entitled "Analysis of fracture deformation field and energy evolution of granite after high confining pressure cyclic load pre-damage" in its current form for publication in Royal Society Open Science.

on behalf of Prof R. Kerry Rowe (Subject Editor)
openscience@royalsociety.org

Appendix A

Dear editor:

Thank you for your useful comments and suggestions on the language and the structure of our manuscript. We have modified the manuscript accordingly, and the detailed corrections are listed below point by point:

Response to comments on “Analysis of fracture deformation field and energy evolution of granite after high confining pressure cyclic load pre-damage (RSOS-201966)”

Reviewer: 1

1、 The third item "loading in the form of sine wave" in the test scheme, the specific parameters are unknown, which has a great impact on the experimental results, so the specific experimental parameters should be given.

Answer: It had been revised according to the reviewer's opinions. Experimental parameter setting is added

2、 Fig 1 is not clear, please make it larger – It is a little bit hard for me to pick up key information.

Answer: It had been revised according to the reviewer's opinions. The picture is processed clearly and enlarged.

3、 In the method section, it is unclear what is stress level applied on your sample in the cyclic load.

Answer: In order to keep the rock from being damaged during the cycle, the uniaxial compressive strength of rock is chosen as the axial loading stress, and it is also discussed in detail.

4、 Please modify the format of the formula in the text, such as in Formula 1 and 2 need to be italicized uniformly.

Answer: It had been revised according to the reviewer's opinions.

Reviewer: 2

1. In the section ‘2.1 High confining pressure cyclic load pre-damage test’, the author is requested to complete the data for Groups A2, A3, A4 and B in the format of Group A1 in Table 1.

Answer: It had been revised according to the reviewer's opinions. Experimental data are added in Table 1.

2. In the section ‘3.1 Analysis of the pre-damage degree of high confining pressure cyclic load’, as can be seen in Table 2, the degree of damage of rock increases as the confining pressure increases from 50mpa to 80mpa; while the degree of damage of rock decreases as the confining pressure increases from 80mpa to 100mpa; and the degree of damage of rock remains essentially unchanged as the pressure increases from 100mpa to 120mpa, for which the authors are asked to briefly explain.

Answer: It had been revised according to the reviewer's opinions. Under the experimental conditions, when the confining pressure is less than 80 MPa, the confining pressure weakens the strength of

rock and increases the damage. When the confining pressure is greater than 80 MPa, the confining pressure strengthens the damage of rock and decreases the damage.

3. In the section '3.2 Strain evolution during cyclic loading', from the trend of the change of the ratio of elastic strain to strain under different confining pressure in Figure 7, it can be seen that when the confining pressure is 50Mpa, 80Mpa and 100Mpa separately, the ratio of circumferential elastic strain to circumferential strain has a more obvious linear relationship; however, when the confining pressure is increased to 120Mpa, the ratio of circumferential elastic strain to circumferential strain no longer has a linear relationship and fluctuates greatly, does this indicate that the rock has already undergone macroscopic breakage at this time, please briefly explain by the author.

Answer: This paper mainly studies the damage state of rock under the condition of deep complex in-situ stress and the mechanical properties of rock after damage. Therefore, in order to avoid macroscopic failure of rock in the process of damage, axial loading stress is chosen as the uniaxial compressive strength of rock. When the confining pressure is increased to 120Mpa, the ratio of circumferential elastic strain to circumferential strain no longer has a linear relationship and fluctuates greatly, but the rock has not undergone macroscopic breakage at this time. The main reason may be that the confining pressure is too large, which causes the dilatancy of rock and has a great influence on the circumferential strain.

4. In the section '3.4 Dislocation of the pre-damaged specimen', Figure 12 plots the deformation energy density evolution curve, which shows the process of energy release and energy accumulation of the specimen. When the energy accumulation reaches a maximum, a sudden release of energy occurs and the specimen is not undergoing energy accumulation, does it imply that the rock mass no longer has a bearing capacity at this point.

Answer: When the energy accumulation reaches a maximum, a sudden release of energy occurs and the specimen is not undergoing energy accumulation, it imply that the rock mass no longer has a bearing capacity. Because the energy accumulation and release is a dynamic process in the process of rock loading, when the energy accumulation reaches the maximum value, it will exceed the maximum bearing capacity of rock and eventually lead to rock failure.

5. There are many verbose expressions in the paper, so the article content should be modified.

Answer: It had been revised according to the reviewer's opinions.